# Phase 1b trial of anti-EGFR antibody JMT101 and Osimertinib in EGFR exon 20 insertion-positive non-small-cell lung cancer

Shen Zhao[1,20], Wu Zhuang[2,20], Baohui Han[3,20], Zhengbo Song[4,20], Wei Guo ⬡[5,20], Feng Luo[6,20], Lin Wu[7], Yi Hu[8], Huijuan Wang[9], Xiaorong Dong[10], Da Jiang[11], Mingxia Wang[12], Liyun Miao[13], Qian Wang[14], Junping Zhang[15], Zhenming Fu[16], Yihua Huang[1], Chunwei Xu[17], Longyu Hu[18], Lei Li[19], Rong Hu[19], Yang Yang[19], Mengke Li[19], Xiugao Yang[19,21] ✉, Li Zhang[1,21] ✉, Yan Huang[1,21] ✉ & Wenfeng Fang[1,21] ✉

EGFR exon 20 insertion (20ins)-positive non-small-cell lung cancer (NSCLC) is an uncommon disease with limited therapeutic options and dismal prognosis. Here we report the activity, tolerability, potential mechanisms of response and resistance for dual targeting EGFR 20ins with JMT101 (anti-EGFR monoclonal antibody) plus osimertinib from preclinical models and an open label, multi-center phase 1b trial (NCT04448379). Primary endpoint of the trial is tolerability. Secondary endpoints include objective response rate, duration of response, disease control rate, progression free survival, overall survival, the pharmacokinetic profile of JMT101, occurrence of anti-drug antibodies and correlation between biomarkers and clinical outcomes. A total of 121 patients are enrolled to receive JMT101 plus osimertinib 160 mg. The most common adverse events are rash (76.9%) and diarrhea (63.6%). The confirmed objective response rate is 36.4%. Median progression-free survival is 8.2 months. Median duration of response is unreached. Subgroup analyses were performed by clinicopathological features and prior treatments. In patients with platinum-refractory diseases ($n = 53$), confirmed objective response rate is 34.0%, median progression-free survival is 9.2 months and median duration of response is 13.3 months. Responses are observed in distinct 20ins variants and intracranial lesions. Intracranial disease control rate is 87.5%. Confirmed intracranial objective response rate is 25%.

As the third most common activating mutations in epidermal growth factor receptor (EGFR), exon 20 insertions (20ins) comprise 10-12% cases in EGFR-addicted non-small-cell lung cancer (NSCLC)[1–3]. However, unlike classic EGFR mutations that benefit from tyrosine kinase inhibitors (TKIs), EGFR 20ins are resistant to early generations of EGFR TKIs and have been deemed untargetable until recently.

EGFR 20ins represent a heterogenous group of variants characterized by in-frame insertions of 1 to 7 amino acids in the αC-helix or αC-β4 loop of the EGFR tyrosine kinase domain (TKD)[4]. To date, more than 60 distinct EGFR 20ins variants have been identified[5]. Each harboured unique structural and conformational features in their TKDs, while similar ATP-binding pockets resembling the EGFR wildtype.

Hence, the development of an effective 20ins-directed TKI that could cover a wide spectrum of variants while maintaining wildtype selectivity is challenging. Recently, amivantamab, a nonmutation-specific, EGFR-MET bispecific antibody, became the first FDA-approved targeted drug for this population. It provided a confirmed overall response rate (ORR) of 40% and a median progression-free survival (PFS) of 8.3 months in the CHRYSALIS study[6]. Mobocertinib, an EGFR TKI yielding a confirmed ORR of 28% and a median PFS of 7.3 months in a pivotal phase 2 study, became the second targeted drug approved for this population[7]. However, similar to other TKIs, the efficacy of mobocertinib is restricted by the structural heterogeneity of 20ins TKDs. Although a median duration of response (DOR) of 17.5 months demonstrates sustained clinical benefits in responding patients, an ORR of 28% is still below our expectation for a targeted therapy. Meanwhile, the central nervous system (CNS) activity of currently approved drugs are unclear. The CHRYSALIS study and the pivotal study of mobocertinib both excluded patients with untreated brain metastasis or any leptomeningeal diseases[6,7]. In the CHRYSALIS study, 31.6% of patients with treated brain metastasis still developed intracranial progression on amivantamab[8]. Mobocertinib led to a confirmed ORR of 18% in patients with baseline brain metastases[7]. Therefore, there is still room for improvement in 20ins-targeted therapies in terms of overall efficacy, wildtype selectivity and intracranial activity.

Currently, there are a number of 20ins-directed therapies in development[5]. Previously, we reported that dual targeting EGFR with a monoclonal antibody (cetuximab) and TKI (afatinib or osimertinib) led to sustained tumor control in patients with EGFR 20ins-positive NSCLC[9,10]. Preclinical studies by Hasegawa et al. and clinical cases from van Veggel et al. reported similar results[11,12]. JMT101 is an anti-EGFR IgG1 monoclonal antibody developed using cetuximab as a prototype. It shared a similar backbone with cetuximab. Through glycosylation modification, humanization and affinity maturation, JMT101 had reduced immunogenicity, less likelihood of cross-reactivity and a six-fold increase in target affinity in comparison to cetuximab. It demonstrated desirable antitumor activity in vitro and in vivo. In the first-in-human study (NCT04689100)[13], JMT101 monotherapy and in combination with chemotherapy both showed favorable pharmacokinetic properties and safety profiles in patients with advanced colorectal cancers.

In this work, we report the antitumor activity, tolerability, potential mechanisms of response and resistance for dual targeting EGFR 20ins with JMT101 plus afatinib or osimertinib from preclinical models and a phase 1b clinical trial (NCT04448379). Our results demonstrate that JMT101 plus osimertinib has the potential to become a new treatment option for EGFR 20ins-positive NSCLC. Acquired resistance to the combination was predominantly driven by EGFR-independent mechanisms that partly overlapped with those observed in classic EGFR mutations.

## Results

### JMT101 plus afatinib or osimertinib potently inhibit EGFR 20ins in vitro and in vivo

To assess the antitumor activity of JMT101 plus afatinib or osimertinib, common EGFR 20ins in NSCLC, A767_V769dup (insASV), S768_D770dup (insSVD) and N771_H773dup (insNPH), were stably expressed in Ba/F3 cells. Treatment with JMT101 alone from 1 to 200ug/ml had minimal effect on cell viability (Supplementary Fig 1a). While adding JMT101 (10ug/ml) to afatinib or osimertinib significantly shift the dose-response curves to the left and demonstrated potent antiproliferative effects (Supplementary Fig 1b). In xenograft models carrying EGFR insASV (8 mice/group), afatinib or osimertinib alone showed limited antitumor activity (Supplementary Fig 1c, e). Combination treatment with JMT101 50 mg/kg biweekly plus osimertinib 25 mg/kg daily induced an average 44% tumor shrinkage relative to the

pretreatment tumor size on day 14 ($P < 0.001$, Supplementary Fig 1e). The tumor growth inhibition (TGI) index was 103%. In the group treated with JMT101 50 mg/kg biweekly plus afatinib 15 mg/kg daily, significant inhibition of tumor growth was also observed (TGI = 89%, $P < 0.001$, Supplementary Fig 1c). In comparison to afatinib monotherapy or osimertinib monotherapy, the addition of JMT101 did not lead to significant weight loss or treatment-related mortality (Supplementary Fig 1d). Of note, although JMT101 monotherapy showed minimal in vitro activity in EGFR 20ins, treatment with JMT101 in xenograft models led to a 60% tumor growth inhibition ($P < 0.001$). In EGFR 20ins cells (insASV, insSVD, insNPH) cocultured with natural killer cells (effector: target=4:1), JMT101 induced cytotoxicity in a dose-dependent manner (Supplementary Fig 1f). These findings suggest that the antitumor activity of JMT101 may require the participation of effector cells.

### JMT101 plus osimertinib led to a thorough and sustained EGFR blockade

To probe the mechanism of action behind the combination treatment, we first evaluated their impacts on EGFR signaling in 20ins cell lines. Immunoblot analysis was performed on Ba/F3 cells expressing EGFR insASV, insSVD and insNPH. Combining JMT101 with afatinib or osimertinib strongly inhibited EGFR signaling activation in three cell lines (Supplementary Fig 2a). In contrast, neither JMT101 nor osimertinib alone could efficiently block EGFR signaling activation in cells expressing EGFR 20ins (Supplementary Fig 2a). Afatinib markedly inhibited phosphorylation of EGFR, but failed to block downstream components of EGFR pathways, which may explain its lack of activity in EGFR 20ins.

We noticed that the addition of JMT101 reduced total EGFR levels in three cell lines. To visualize these impacts, we performed immunofluorescence staining on Ba/F3 cells expressing EGFR insASV under specific treatments. Compared with cells treated with afatinib or osimertinib alone, cells receiving the combination therapy exhibited a marked downregulation of total EGFR levels and a trend of EGFR internalization (Supplementary Fig 2b). Flow cytometry was conducted to further investigate the impacts of JMT101 plus afatinib or osimertinib on EGFR distribution (Supplementary Fig 3). In Ba/F3 cells expressing EGFR insASV, treatment with afatinib or osimertinib significantly increase the level of cell surface EGFR, while the addition of JMT101 led to a marked reduction in cell surface EGFR levels. Particularly, the level of cell surface EGFR decreased even further after 24 h of treatment with JMT101 plus osimertinib (Supplementary Fig 3). Taken together, dual targeting EGFR 20ins with JMT101 plus osimertinib led to a thorough and sustained EGFR blockade via suppressing signaling activation, inducing receptor internalization and downregulation.

### Study design and participants

Between June 29, 2020, and December 28, 2021, 150 patients with EGFR 20ins-positive advanced NSCLC were enrolled into the phase 1b trial from 15 sites in China (Fig. 1). The complete protocol and statistical analysis plan were presented in the Supplementary Note. The primary objective was to evaluate safety and tolerability of JMT101 plus afatinib or osimertinib. Secondary objectives included anti-tumor activity measured by tumor responses, duration of response, progression-free survival and overall survival (OS), pharmacokinetic and immunogenicity profiles, and biomarkers potentially associated with clinical outcomes. Twelve patients were enrolled during the dose-escalation stage, receiving JMT101 6 mg/kg every 2 weeks plus afatinib 30 mg daily (cohort A1, $n = 3$), JMT101 6 mg/kg every 2 weeks plus afatinib 40 mg daily (cohort A2, $n = 3$), JMT101 6 mg/kg every 2 weeks plus osimertinib 80 mg daily (cohort B1, $n = 3$) and JMT101 6 mg/kg every 2 weeks plus osimertinib 160 mg daily (cohort B2, $n = 3$), respectively. Although JMT101 monotherapy was tolerated up to the dose level of 10 mg/kg in its first-in-human study, 6 mg/kg was selected for the

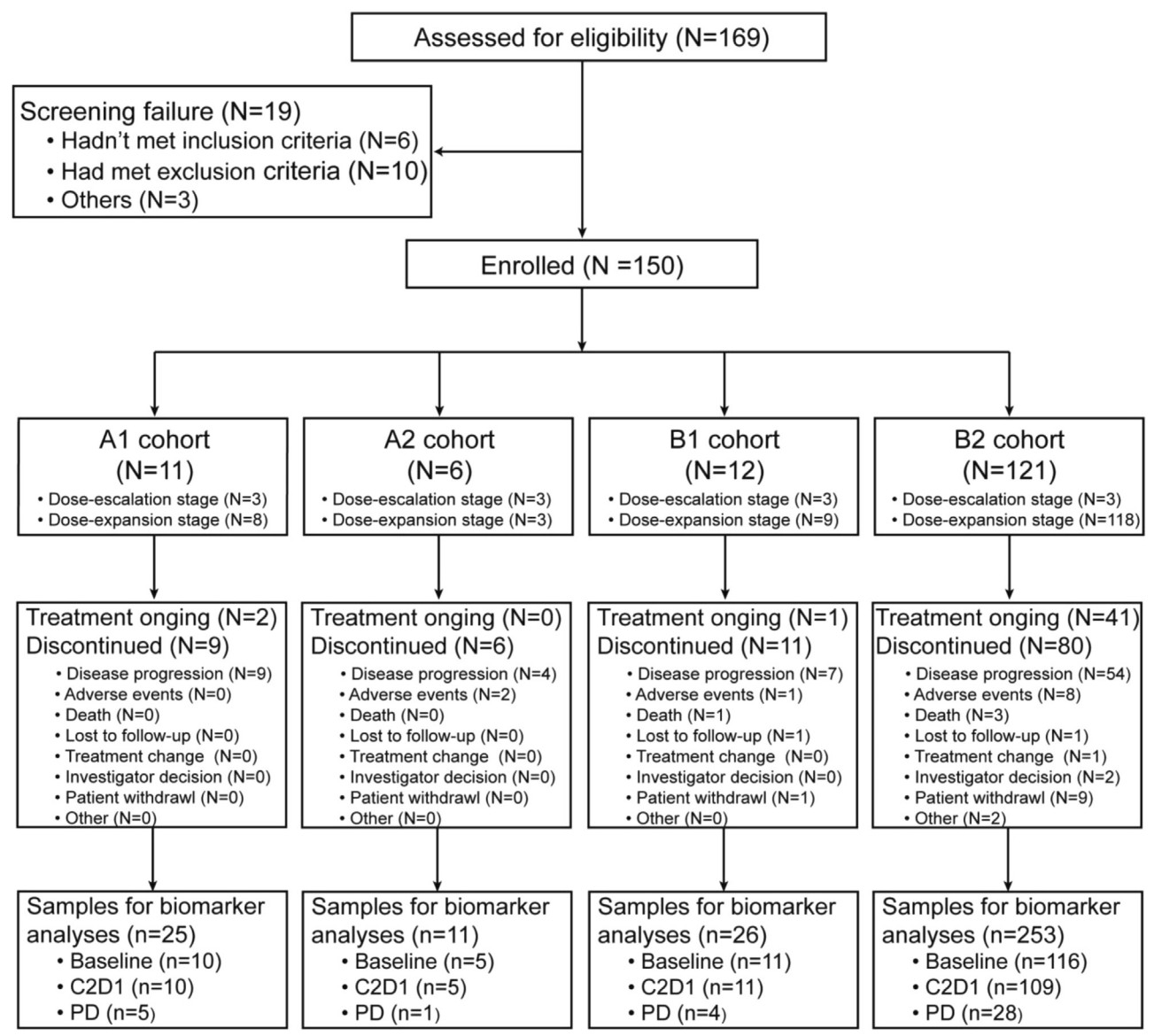

**Fig. 1 | Flow diagram.** Study design and outline of number of patients (N) and samples (n) available for analyses.

combination therapy for similar efficacy and better tolerability in comparison to higher dose levels[13]. A total of 138 patients were enrolled into the dose-expansion stage (cohort A1 = 8, cohort A2 = 3, cohort B1 = 9, cohort B2 = 118) for further evaluation of efficacy and safety. Cohort B2 (JMT101 plus osimertinib 160 mg) was selected for further expansion due to better efficacy-safety profiles observed and higher activity of osimertinib 160 mg over 80 mg shown in previous trials on EGFR 20ins[14–17]. A total of 121 patients were enrolled into cohort B2 (efficacy population).

All patients had EGFR 20ins documented by local testing at enrollment, 143 of whom (95.3%) were further confirmed by central testing on tumor tissue samples ($n = 143$) and/or paired peripheral blood samples ($n = 117$). A total of 38 distinct EGFR 20ins variants were identified (Supplementary Tab 1). The most common 20ins variants were A767_V769dup ($n = 49$, 34.3%), S768_D770dup ($n = 29$, 20.3%), P772_H773dup ($n = 10$, 7.0%), H773dup ($n = 7$, 4.9%) and N771_H773dup ($n = 7$, 4.9%). According to the relationship between insertion location and the αC-helix, 120 patients (83.9%) carried near-loop insertions (A767-P772), 18 patients (12.6%) carried far-loop insertions (H773-C775), and five patients (3.5%) had helical insertions. Baseline characteristics of patients were presented in Table 1. In the efficacy

population, the median age was 60 years (range, 29–77). Most patients were female ($n = 61$, 50.4%), never smokers ($n = 78$, 64.5%) and had an ECOG performance status of 1 ($n = 106$, 87.6%). More than half of the efficacy population ($n = 62$, 51.2%) had baseline CNS metastases. Among them, 80.6% ($n = 50$) were untreated, and 12 patients had received brain radiotherapy. Three patients had untreated leptomeningeal metastases. The median number of previous systemic therapy was 1 (range, 0–7). Sixteen patients had been treated with EGFR TKIs, including osimertinib (80 mg $n = 5$, 160 mg $n = 1$), afatinib ($n = 4$), almonertinib ($n = 3$), gefitinib ($n = 3$), poziotinib ($n = 1$), furmonertinib ($n = 1$) and icotinib ($n = 1$). By data cutoff on May 31, 2022, the median follow-up for the safety population ($n = 150$) and efficacy population ($n = 121$) was 9.8 months (range, 1.1–22.8 months) and 9.1 months (range, 1.1–18.9 months), respectively.

## Safety
Primary objective of this trial is to evaluate the safety and tolerability of JMT101 plus afatinib or osimertinib in advanced NSCLC with EGFR 20ins. The overall study population ($n = 150$) was included in safety analysis. No dose-limiting toxicity was observed in the dose-escalation stage. Across all cohorts, treatment-emergent adverse events (TEAEs)

**Table 1 | Baseline demographics characteristics**

| Characteristics | A1 cohort (n = 11) | A2 cohort (n = 6) | B1 cohort (n = 12) | B2 cohort (n = 121) | All (n = 150) |
|---|---|---|---|---|---|
| Age, median (range, y) | 61 (41, 65) | 50 (33, 67) | 59 (26, 71) | 60 (29, 77) | 60 (26, 77) |
| Sex, no. (%) | | | | | |
| Female | 9 (81.8) | 2 (33.3) | 10 (83.3) | 61 (50.4) | 82 (54.7) |
| Male | 2 (18.2) | 4 (66.7) | 2 (16.7) | 60 (49.6) | 68 (45.3) |
| Histologic type, no. (%) | | | | | |
| Adenocarcinoma | 11 (100) | 6 (100) | 12 (100) | 119 (98.3) | 148 (98.7) |
| Squamous cell carcinoma | 0 | 0 | 0 | 1 (0.8) | 1 (0.7) |
| Adenosquamous | 0 | 0 | 0 | 1 (0.8) | 1 (0.7) |
| ECOG performance status, no. (%) | | | | | |
| 0 | 4 (36.4) | 2 (33.3) | 2 (16.7) | 15 (12.4) | 23 (15.3) |
| 1 | 7 (63.6) | 4 (66.7) | 10 (83.3) | 106 (87.6) | 127 (84.7) |
| History of smoking, no. (%) | | | | | |
| Never | 9 (81.8) | 4 (66.7) | 11 (91.7) | 78 (64.5) | 102 (68.0) |
| Ever | 2 (18.2) | 2 (33.3) | 1 (8.3) | 43 (35.5) | 48 (32.0) |
| Disease stage, no. (%) | | | | | |
| IV | 10 (90.9) | 6 (100) | 11 (91.7) | 119 (98.3) | 146 (97.3) |
| IIIB | 0 | 0 | 1 (8.3) | 0 | 1 (0.7) |
| Prior systemic therapies | | | | | |
| Median (range) | 1 (0, 6) | 1 (1, 3) | 1 (0, 6) | 1 (0, 7) | 1 (0, 7) |
| Prior systemic anticancer regimens, no. (%) | | | | | |
| Chemotherapy | 10 (91.9) | 5 (83.3) | 7 (58.3) | 63 (52.1) | 85 (56.7) |
| EGFR TKI | 1 (9.1) | 1 (16.7) | 5 (41.7) | 16 (13.2) | 23 (15.3) |
| Baseline CNS metastases, no. (%) | | | | | |
| Yes | 4 (36.4) | 3 (50.0) | 6 (50.0) | 62 (51.2) | 75 (50.0) |
| No | 7 (63.6) | 3 (50.0) | 6 (50.0) | 59 (48.8) | 75 (50.0) |

*CNS* central nervous system, *ECOG* Eastern Cooperative Oncology Group, *EGFR* epidermal growth factor receptor, *TKI* tyrosine kinase inhibitor.

**Table 2 | Incidence of all-grade TRAEs ≥ 10% and grade 3 or higher TRAEs ≥ 3%**

| TEAE, n (%) | Safety population (n = 150) | | B2 cohort (n = 121) | |
|---|---|---|---|---|
| | All grade | Grade ≥ 3 | All grade | Grade ≥ 3 |
| Any | 147 (98.0) | 93 (62.0) | 119 (98.3) | 79 (65.3) |
| Rash | 118 (78.7) | 32 (21.3) | 96 (79.3) | 26 (21.5) |
| Diarrhea | 98 (65.3) | 16 (10.7) | 80 (66.1) | 14 (11.6) |
| Dry skin | 88 (58.7) | 7 (4.7) | 67 (55.4) | 5 (4.1) |
| Decreased appetite | 85 (56.7) | 4 (2.7) | 67 (55.4) | 3 (2.5) |
| Paronychia | 81 (54.0) | 3 (2.0) | 64 (52.9) | 2 (1.7) |
| AST increased | 57 (38.0) | 2 (1.3) | 44 (36.4) | 2 (1.7) |
| ALT increased | 54 (36.0) | 0 | 44 (36.4) | 0 |
| Weight decreased | 52 (34.7) | 6 (4.0) | 50 (41.3) | 6 (5.0) |
| Vomiting | 50 (33.3) | 3 (2.0) | 39 (32.2) | 3 (2.5) |
| Oral mucositis | 47 (31.3) | 4 (2.7) | 37 (30.6) | 3 (2.5) |
| Nausea | 43 (28.7) | 1 (0.8) | 36 (29.8) | 1 (0.7) |
| Pruritus | 42 (28.0) | 2 (1.3) | 29 (24.0) | 2 (1.7) |
| Anemia | 40 (26.7) | 5 (3.3) | 33 (27.3) | 4 (3.3) |
| Malaise | 39 (26.0) | 5 (3.3) | 29 (24.0) | 5 (4.1) |
| White blood cell decreased | 37 (24.7) | 4 (2.7) | 33 (27.3) | 4 (3.3) |
| Platelet count decreased | 36 (24.0) | 7 (4.7) | 35 (28.9) | 6 (5.0) |
| Hypoalbuminemia | 36 (24.0) | 0 | 28 (23.1) | 0 |
| Neutrophil count decreased | 29 (19.3) | 8 (5.3) | 28 (23.1) | 8 (6.6) |
| Fever | 29 (19.3) | 0 | 21 (17.4) | 0 |
| Skin hypopigmentation | 29 (19.3) | 0 | 21 (17.4) | 0 |
| Dizziness | 26 (17.3) | 0 | 18 (14.9) | 0 |
| Hyponatremia | 24 (16.0) | 1 (0.7) | 20 (16.5) | 1 (0.8) |
| Proteinuria | 22 (14.7) | 0 | 21 (17.4) | 0 |
| Stomal ulcer | 22 (14.7) | 0 | 20 (16.5) | 0 |
| Hypokalemia | 21 (14.0) | 2 (1.3) | 10 (8.3) | 0 |
| CPK increased | 20 (13.3) | 5 (3.3) | 20 (16.5) | 5 (4.1) |
| Headache | 19 (12.7) | 1 (0.7) | 12 (9.9) | 1 (0.8) |
| Skin fissures | 18 (12.0) | 0 | 16 (13.2) | 0 |
| QT interval prolonged | 16 (10.7) | 0 | 16 (13.2) | 0 |
| Blood LDH increased | 16 (10.7) | 0 | 14 (11.6) | 0 |
| Blood creatinine increased | 16 (10.7) | 0 | 13 (10.7) | 0 |
| Hypocalcemia | 15 (10.0) | 0 | 14 (11.6) | 0 |

*TRAEs* treatment-related adverse events, *AST* aspartate aminotransferase, *ALT* alanine aminotransferase, *CPK* creatine phosphate kinase, *LDH* lactate dehydrogenase.

of any grade occurred in 149 out of 150 patients (99.3%), and were considered as treatment-related in 147 patients (98%). The most common all-grade treatment-related adverse events with incidence ≥50% were rash (n = 118, 78.7%), diarrhea (n = 98, 65.3%), dry skin (n = 88, 58.7%), decreased appetite (n = 85, 56.7%) and paronychia (n = 81, 54.0%). Most events were grade 1 or 2. Infusion-related reactions (n = 7, 4.7%) and interstitial lung diseases (n = 2, 1.3%) were uncommon. Grade ≥3 treatment-related adverse events were observed in 62.0% patients (n = 93). Grade ≥3 events that occurred in ≥10% of patients were rash (n = 32, 21.3%) and diarrhea (n = 16, 10.7%). Common treatment-related adverse events (≥10%) and grade ≥3 events ((≥3%) are listed in Table 2. Adverse events in dose-escalating cohorts are listed in Supplementary Tab 2.

All events were resolved with supportive care, dose interruption, reduction, and/or treatment discontinuation. Adverse events leading to dose interruption and dose reduction of any drug were documented in 57.3% (n = 86) and 28.0% (n = 42) of patients, respectively. The most common AEs leading to dose reduction (n ≥ 5) were rash (n = 22, 14.7%) and dry skin (n = 5, 3.3%). Treatment-related discontinuation occurred in 4.0% (n = 6) of patients due to rash (n = 2), pyrexia (n = 1), interstitial lung disease (n = 1), nephrotic syndrome (n = 1) and deep venous thromboembolism in lower extremities (n = 1). No treatment-related grade 5 adverse event was observed.

**Pharmacokinetics and immunogenicity**
Secondary objectives of the phase 1b trial include anti-tumor activity, pharmacokinetics, immunogenicity profiles, and potential biomarkers.

Plasma samples from patients in cohort A1, A2, B1, and B2 were obtained for pharmacokinetic analysis. A summary of the PK parameters is shown in Supplementary Tab 3 and Supplementary Fig 4. Overall, the pharmacokinetic parameters of JMT101 in combination with afatinib or osimertinib were similar to those reported in previous phase 1 trial where JMT101 was used as a monotherapy[13]. JMT101 had reached steady state by the third dose and exhibited linear pharmacokinetics at the dose of 6 mg/kg. Oral administration of afatinib or osimertinib did not change the pharmacokinetic profile of JMT101. The pharmacokinetic parameters of JMT101 were similar between the four cohorts.

The incidence of antibodies to JMT101 was low (18/132, 13.6%). Among all positive samples, three samples had antibody titer greater than 10. No patients had persistent antibody positivity. Neither antibody positivity nor antibody titer exerted evident impact on pharmacokinetics, efficacy, and safety of JMT101.

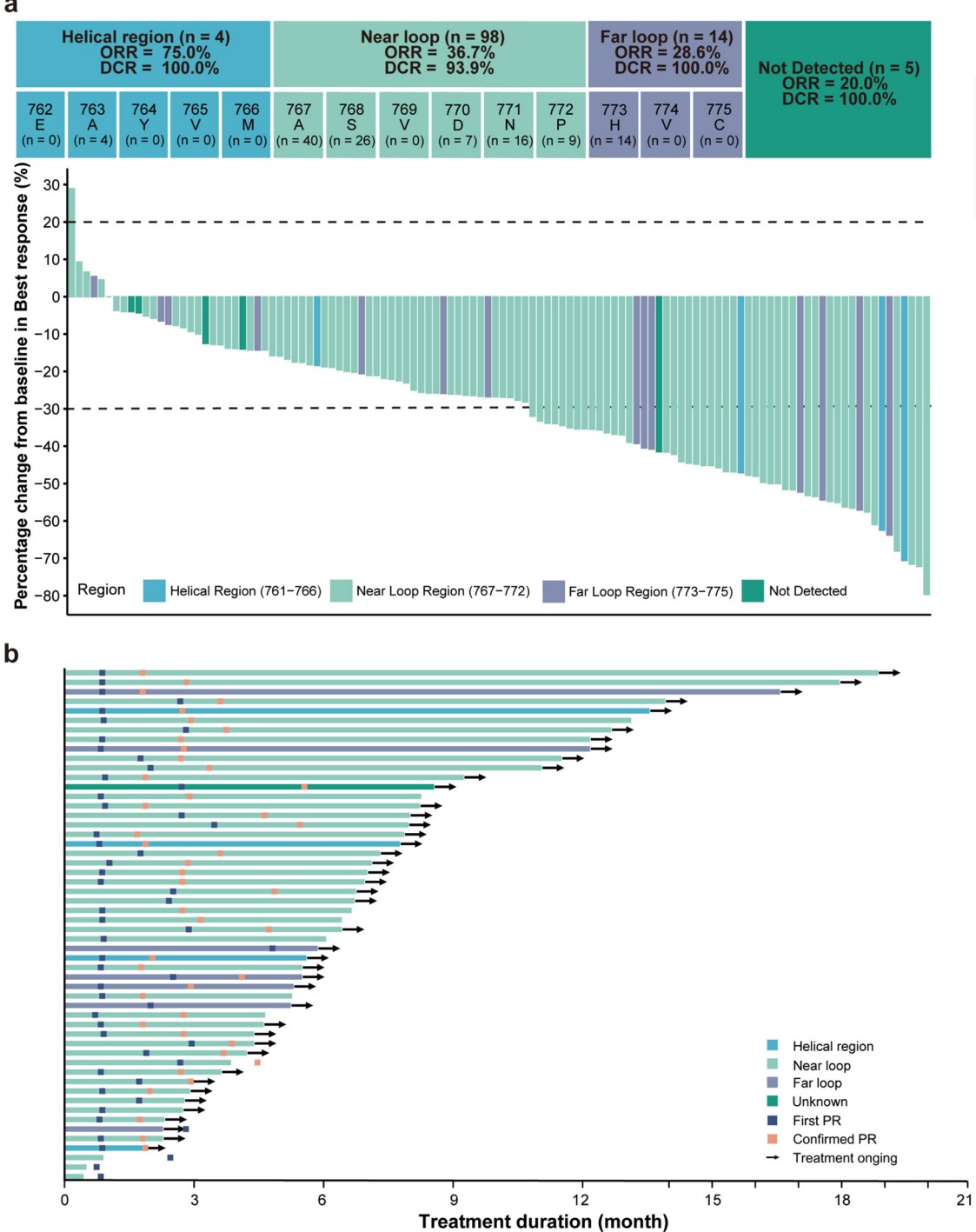

**Fig. 2 | Response characteristics in the efficacy population. a** Maximal percentage change in the sum of target lesions from baseline based on IRC assessment (*n* = 116). Five patients carrying near-loop insertions were deemed unevaluable by the IRC and are not included in the plot. Dotted lines at 20% and −30% indicate cutoffs for progressive disease and partial response per RECIST v1.1, respectively; **b** Time to response and duration of response in patients with confirmed and unconfirmed responses to treatment per IRC assessment (*n* = 54).

## Radiological efficacy

Antitumor efficacy is one of the secondary outcomes of the clinical trial. Confirmed partial responses were observed in 2 out of 11 patients from cohort A1 (18%), 2 out of 6 patients from cohort A2 (33%), and 5 out of 12 patients from cohort B1 (42%, Supplementary Fig. 5). In the efficacy population (*n* = 121), 91.7% of patients (*n* = 111) experienced tumor shrinkage at the first radiological review based on independent review committee (IRC) assessment (Fig. 2a). Objective tumor

responses per IRC were documented in 54 patients, providing an objective response rate (ORR) of 44.6% (54/121, 95% CI = 35.6–53.9, Fig. 2a). Among them, 44 patients had confirmed responses and 10 patients had stable disease as the best of response. The confirmed objective response rate and disease control rate (DCR) based on IRC assessment was 36.4% (44/121, 95% CI = 27.8–45.6) and 95.0% (115/121, 95% CI = 89.5–98.2), respectively (Fig. 2a). Efficacy outcomes assessed by investigators were consistent with IRC (Supplementary Tab 4).

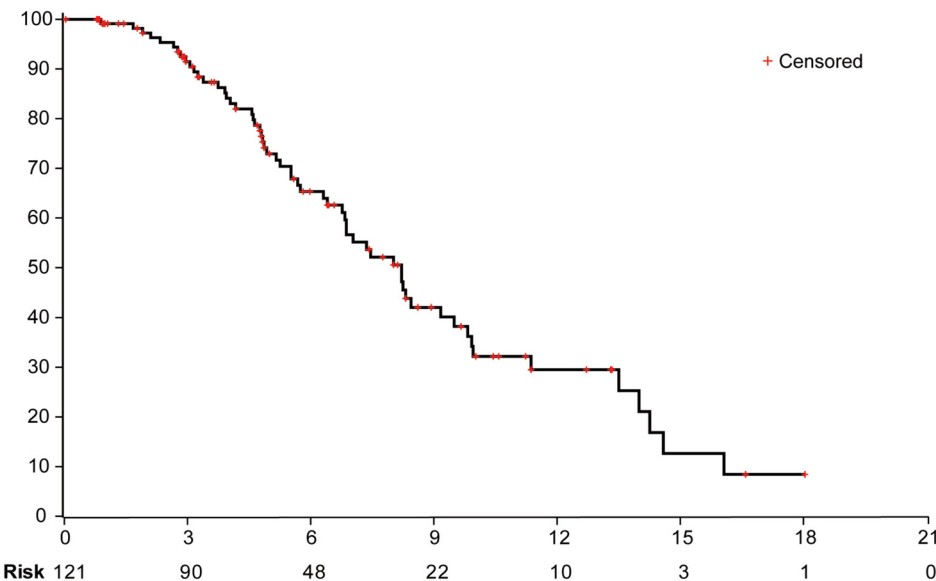

**Fig. 3 | Progression-free survival in the efficacy population.** Kaplan-Meier estimates of progression-free survival in patients treated by JMT101 plus osimertinib (efficacy population) based on IRC assessment.

Non-prespecified, exploratory subgroup analyses were performed by clinicopathological features and prior treatments. Objective tumor responses were observed in patients with different 20ins variants, prior EGFR TKI treatments and untreated CNS metastases. In patients carrying helical, near-loop and far-loop insertions, confirmed objective response rates were 75% (3/4, 95% CI = 19.4–99.4), 36.7% (36/98, 95% CI = 27.2–47.1) and 28.6% (4/14, 95% CI = 8.4–58.1), respectively (Fig. 2a, $P = 0.297$). For patients in the first-line and latter-line settings, confirmed objective response rates were 42.3% (22/52, 95% CI = 28.7–56.8) and 31.9% (22/69, 95% CI = 21.2–42.2%), respectively ($P = 0.238$). Partial responses were observed in patients who progressed on icotinib, afatinib, osimertinib (80 mg qd) and almonertinib. One patient previously treated by poziotinib had a 25% decrease in overall tumor burden and a progression-free survival of 9.8 months. In the subset of patients who progressed on platinum-based chemotherapy and received the investigated treatment in the second-line setting ($n = 53$), JMT101 plus osimertinib led to an IRC-assessed confirmed objective response rate of 34.0% (18/53, 95% CI = 21.5–48.3), disease control rate of 96.2% (51/53, 95% CI = 87.0–99.5) and a median progression-free survival of 9.2 months (95% CI = 5.5–14.3). The median duration of response was reached in this subset of patients (13.3 months, 95% CI = 3.9-not reached). In patients with baseline CNS metastases ($n = 62$), the IRC-assessed confirmed objective response rate and disease control rate were 33.9% (21/62, 95% CI = 22.3–47.0) and 95.2% (59/62, 95% CI = 86.5–99.0), respectively. Sixteen patients had brain metastasis as target lesions. Among them, 13 patients had intracranial tumor shrinkage of any quantity (Supplementary Fig. 6a). The intracranial disease control rate was 87.5% (14/16, 95% CI = 69.3–100). Intracranial complete response was observed in one patient who progressed on poziotinib. Intracranial partial responses were observed in 6 patients and confirmed in 3 patients. Overall, the confirmed intracranial objective response rate was 25.0% (4/16, 1 complete response, 3 partial responses). Three patients had untreated leptomeningeal diseases at baseline. JMT101 plus osimertinib led to 1 partial response (PFS = 14.2 m), 1 stable disease (PFS = 8.0 m) and 1 progressive disease (PFS = 3.2 m) in these patients, respectively.

As of May 31, 2022, 41 patients (33.9%) in efficacy population remained progression-free and stayed on treatment. With a median follow-up of 9.1 months, responses were ongoing in 54.5% of patients with confirmed responses ($n = 24$, Fig. 2b). Events of IRC-assessed

disease progression or death occurred in 58 patients (47.9%). The estimated median progression-free survival was 8.2 months (95% CI = 6.8–9.5, Fig. 3). Median progression-free survival was similar in patients with different characteristics and different 20ins variants (Supplementary Fig. 6b). The rate of progression-free at 6, 9, and 12 months were 65.3% (95% CI = 54.5–74.2), 42.0% (95% CI = 30.5–53.2) and 29.5% (95% CI = 18.4–41.5), respectively. Overall survival was immature with only 29 deaths (24.0%) documented at data cutoff.

## Biomarker analysis of clinical benefits

Peripheral blood samples before treatment, on treatment (cycle 2 day 1, C2D1) and post treatment (after PD) were obtained from 142, 135 and 38 patients, respectively (Supplementary Fig. 7). Using a targeted next-generation sequencing panel, hotspot alterations in 59 genes were detected in 121 patients at baseline (Fig. 4). EGFR 20ins tended to be mutually exclusive with other known drivers in NSCLC, including EGFR (19del, L858R), ALK, ROS1, RET, HER2, MET, BRAF and KRAS. Ten (8.3%) patients had concomitant EGFR amplifications, and one (0.8%) had concurrent MET amplification. The most common concurrent alterations were detected in TP53 ($n = 60$, 49.6%), PIK3CA ($n = 10$, 8.3%) and RB1 ($n = 6$, 4.96%). Most TP53 alterations (86.7%) occurred in the DNA-binding domain (exon 5–8). In comparison to baseline, tumor mutation landscapes on-treatment and post-treatment were generally unchanged, with the most common concurrent alterations still being TP53, PIK3CA and RB1 (Supplementary Fig. 8). While the mean tumor mutation load based on cfDNA analysis significantly decreased at C2D1 ($0.93 \pm 1.15$ vs $2.01 \pm 1.99$, $P < 0.001$), and increased at progression ($2.01 \pm 1.99$ vs $2.92 \pm 2.19$, $P = 0.024$).

To identify potential biomarkers for clinical benefits, baseline genomic profiles from patients with durable clinical benefits (DCB, defined as patients with CR/PR/SD for at least 6 months) were compared to those without. Concurrent TP53 alteration was found enriched in patients without DCB ($P = 0.040$, Fig. 5a). Patients carrying TP53-altered tumors tended to have shorter PFS than TP53-wildtype (7.0 m [95% CI = 5.0–9.1] vs 8.3 m [6.2–10.4], $P = 0.056$, Supplementary Fig 9). No significant correlation was observed between TP53 status and the best of response ($P = 0.652$), suggesting that the presence of TP53 comutations did not affect the initial treatment efficacy. In comparison to TP53-wildtype tumors, TP53-altered tumors had

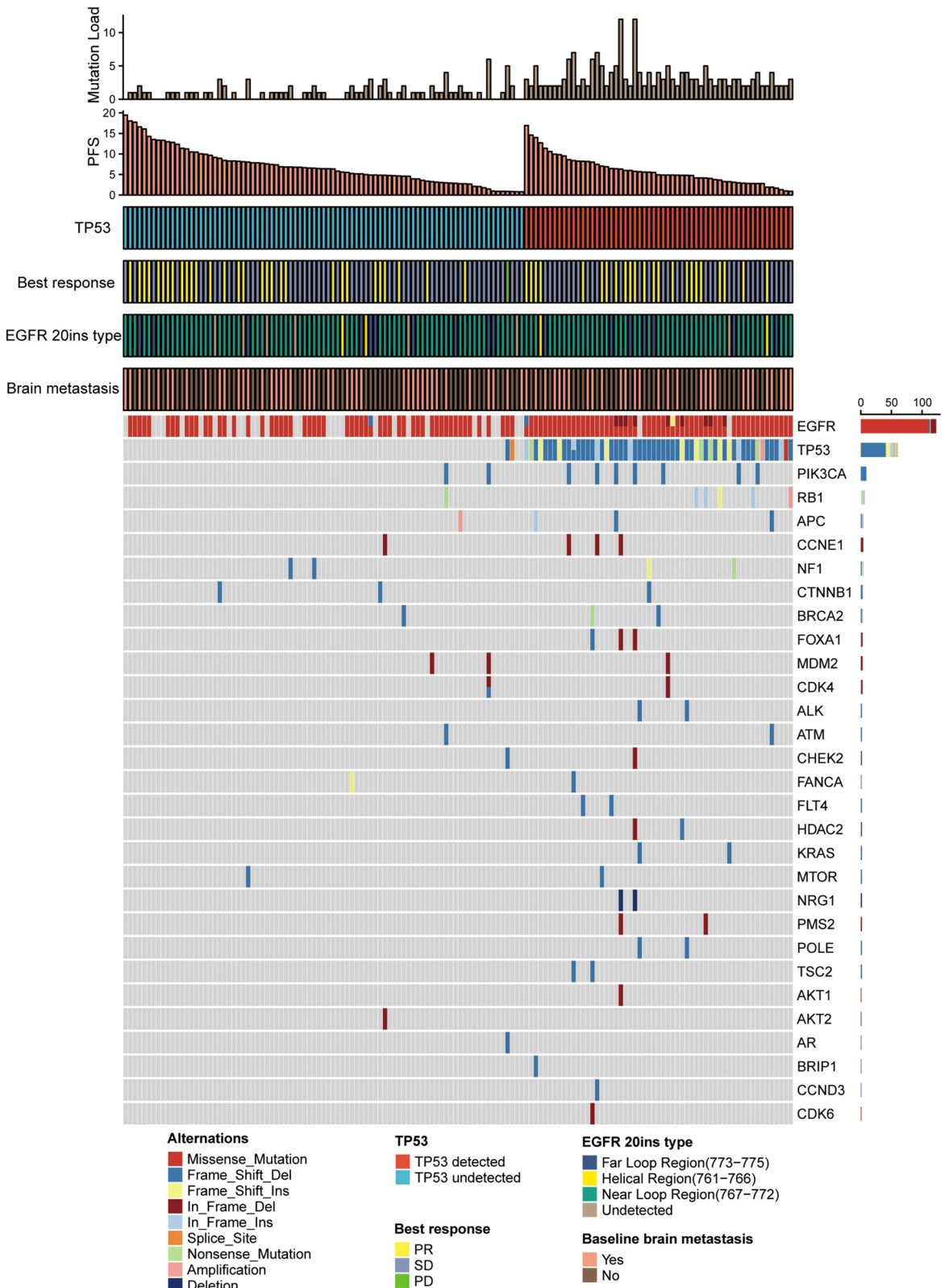

**Fig. 4 | Baseline mutation landscape in patients with EGFR 20ins.** Tumor mutation landscape defined by cfDNA in advanced NSCLC patients with EGFR 20ins at baseline (*n* = 142). Only the top 30 mutated genes are shown here. Source data are provided as a source data file.

significantly higher mutation load at baseline (*P* < 0.001), C2D1 (*P* = 0.018) and disease progression (*P* = 0.001, Supplementary Fig. 9), which may explain early progression in these patients. Further, A lack of durable clinical benefits also correlated with higher tumor mutation load at baseline (*P* = 0.028, Fig. 5b) and at C2D1 (*P* = 0.013, Fig. 5c). Comparing genomic profiles from patients with confirmed responses versus those from others demonstrated that responding tumors had significantly lower mutation load after one month of treatment

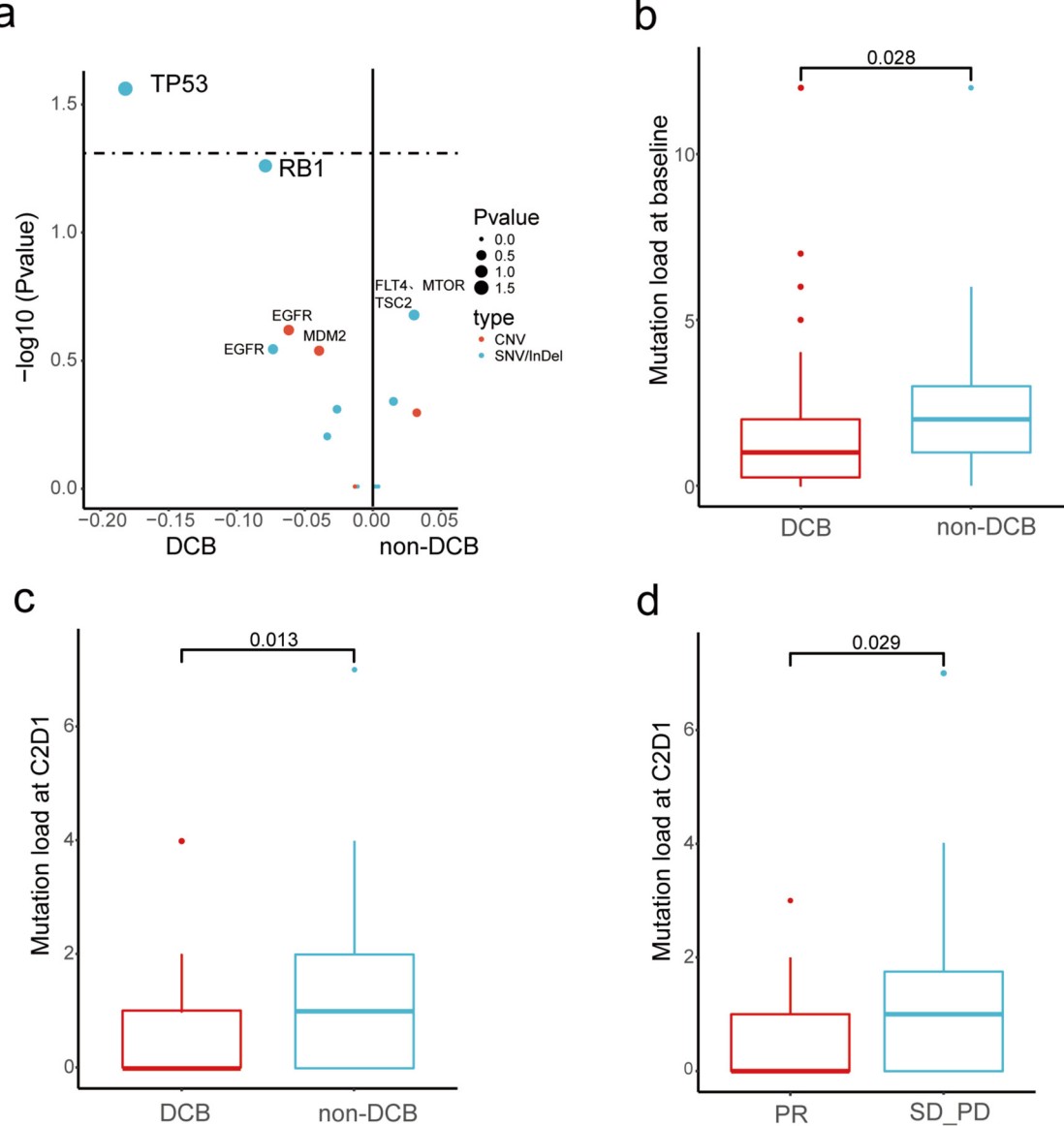

**Fig. 5 | Potential biomarkers for clinical outcomes. a** Bubble plot illustrates the enrichment of baseline gene alterations in patients with early PFS events (<6 m) versus those with late PFS events (≥6 m). The difference of gene alteration frequency between two groups of patients is plotted on the x-axis. The dash line indicated a $P = 0.05$. FLT4, MTOR, and TSC2 shared the same dot as the frequency difference of these genes are the same; $P_{TP53} = 0.028323$, V = 0.184, Chi-square test without adjustment for multiple comparisons was used for statistical analysis, df = 1; Effect size V was measured by Cramer's V. **b, c** Lower tumor mutation load at baseline (Mann-Whitney U Test, two-sided, $P = 0.028$) and C2D1 (Mann-Whitney U Test, two-sided, $P = 0.013$) distinguish patients with early PFS events (<6 m, baseline $n = 76$, C2D1 $n = 70$) from those with late PFS events (≥6 m,

baseline $n = 66$, C2D1 $n = 65$); **d** In comparison to non-responding tumors ($n = 82$), responding tumors ($n = 53$) had lower mutation load after one month of treatment (Mann-Whitney U Test, two-sided, $P = 0.029$). Box-and-whisker plots display box limits, whiskers and outliers, which can be calculated by IQR (Inter Quartile Range, Q3-Q1). The center line in the box plots represents the median; the upper limit of the box plots represents the 75th percentile (Q3); the lower limit of the box plots represents the 25th percentile (Q1); the upper whisker is the maximum value of Q3 + 1.5IQR; and the lower whisker is the minimum value of Q1-1.5IQR. The outlier is defined as a value less than Q1-1.5IQR or greater than Q3 + 1.5IQR. Outliers are showed as point in the box plots. Source data are provided as a source data file.

($P = 0.029$, Fig. 5d). Additionally, we evaluated changes in cfDNA VAF (variant allelic frequency) of EGFR 20ins during treatment (baseline and C2D1, $n = 135$) and their associations with clinical outcomes. Generally, there was a significant decrease in EGFR 20ins VAF on C2D1 ($P < 0.001$, Supplementary Fig. 10a). Patients with responding tumors had greater reduction in VAF in comparison to non-responders ($P = 0.007$, Supplementary Fig. 10b). Meanwhile, cfDNA clearance ($P = 0.012$) and cfDNA decrease (VAF fold change<1) on C2D1 ($P < 0.001$) both significantly correlated with longer PFS in the study population (Supplementary Fig. 10c, d). Taken together, these findings indicate that cfDNA-based tumor mutation load and VAF may serve as

potential biomarkers of responses and clinical benefits for EGFR 20ins-postive NSCLC.

## Potential resistance mechanisms to dual EGFR targeting in EGFR 20ins

To explore potential resistance mechanisms to dual EGFR targeting in EGFR 20ins, serial blood samples before treatment, on treatment (C2D1) and at disease progression from 38 patients were obtained for cfDNA analysis. De novo alterations at progression were found in 86.8% ($n = 33$) of patients. Among them, putative mechanisms that were previously reported as resistance drivers in classic EGFR

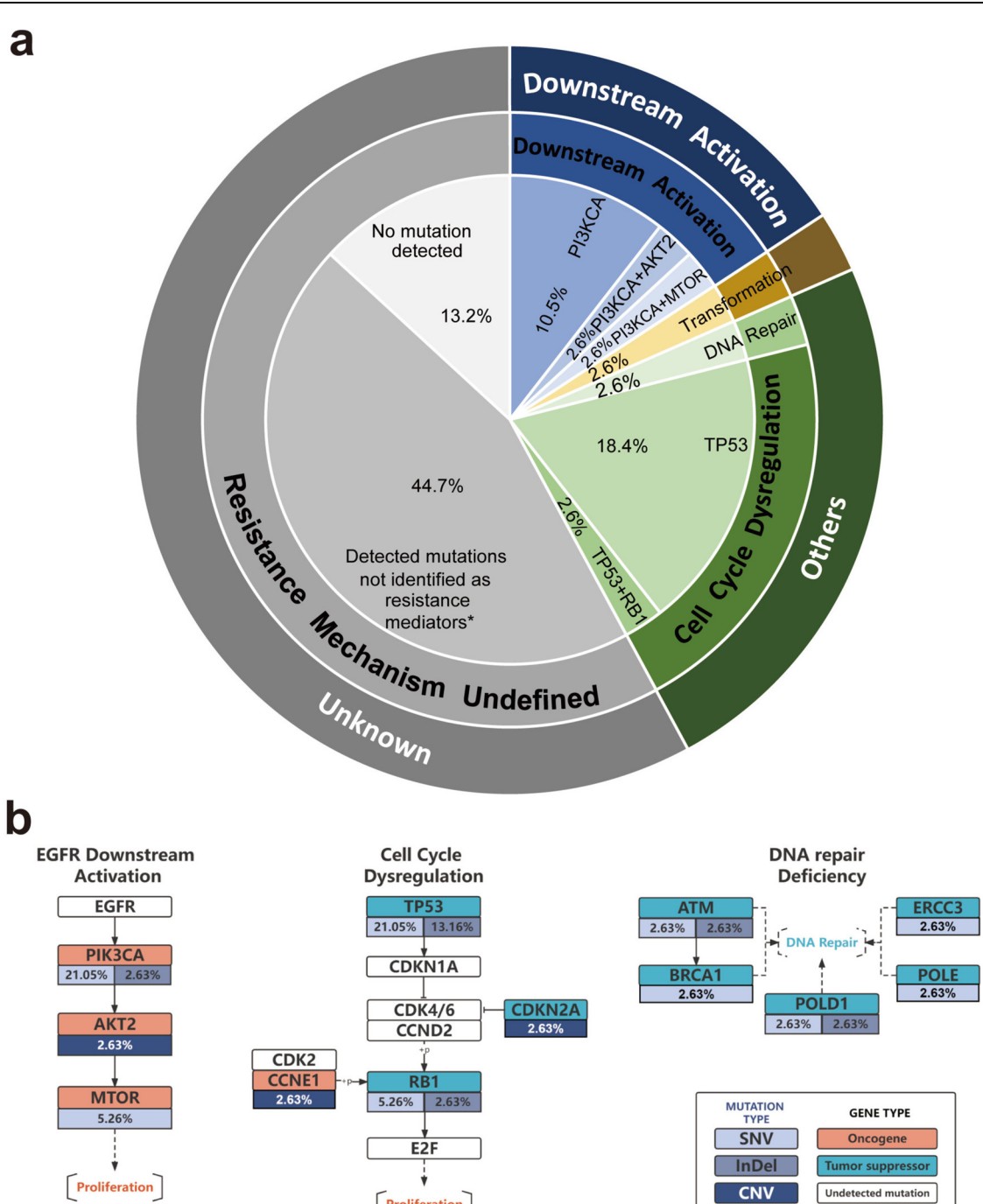

**Fig. 6 | Acquired resistance mechanism to dual EGFR blockade. a** Distribution of established resistance mechanism in patients providing serial blood samples for genomic profiling (n = 38). *Potential resistance mediators were defined as mutations that were newly detected or increased in VAF by 1.5 times at progression and defined as pathogenic mutations by OncoKB[53] and COSMIC database;[54] **b** Potential

mediators of resistance were subjected to KEGG pathway analysis[56]. Altered pathways in patients with downstream signaling activation (n = 6), cell cycle dysregulation (n = 8) and DNA repair defects (n = 1) are detected. Source data are provided as a source data file.

mutations were identified in 42.1% (n = 16) of patients. Interestingly, no acquired EGFR mutation (e.g., C797S) was identified in post-progression samples. Acquired EGFR amplification (copy number gain = 3.4) at progression was only detected in one patient with small-cell transformation. Putative resistance mechanisms identified included signaling activation downstream of EGFR (n = 6, 15.8%), small-cell transformation (n = 1, 2.6%), DNA repair defects (n = 1, 2.6%) and cell cycle dysregulation (n = 8, 21.1%, Fig. 6a).

Among patients with potential drivers for resistance, integrated analysis of matched mutation profiles pre- and post-treatment showed

enriched alterations in genes engaging in the PI3K/AKT pathway, cell cycle regulation and DNA repair (Fig. 6b). The PI3K/AKT pathway had been reported as a resistance driver for TKIs in classic EGFR-mutant NSCLC[18].In patients with EGFR 20ins, acquired alterations were detected in PI3K (SNV 21.1%, Indel 2.6%), AKT2 (2.6%) and MTOR (5.3%). In contrast, alterations in genes involving in cell cycle regulation and DNA repair are not canonical resistance mechanisms to EGFR targeted treatments. These mechanisms were used to be implicated in resistance to chemotherapies rather than to targeted therapies[19]. Their prevalence in the post-progression samples here may correlate with

the addition of JMT101. It should be noted that resistance mechanisms proposed here are merely putative and need to be further validated. But nevertheless, these data suggest that dual EGFR targeting in EGFR 20ins may lead to a unique pattern of resistance centering on EGFR-independent mechanisms.

## Discussion

EGFR 20ins has long been an unadopted orphan in NSCLC. Challenges in developing effective 20ins-targeted therapies lie in the structural heterogeneity of 20ins TKDs and their resemblance to EGFR wildtype[1,20]. Recently, the approval of amivantamab and mobocertinib have marked the dawn of the targeted treatment era for EGFR 20ins-positive NSCLC and set a benchmark for other 20ins-targeted therapies under development. Amivantamab and mobocertinib respectively delivered a confirmed ORR of 40% with a median PFS of 8.3 months, and a confirmed ORR of 28% with a 7.3 months median PFS in this population[6,7]. There is still room for improvement in 20ins-targeted therapies in terms of overall efficacy, wildtype selectivity and intracranial activity. Previously, we found that the combination of EGFR antibody and kinase inhibitor led to sustained tumor control in patients with EGFR 20ins-positive NSCLC[9,10]. Here, we report results from a phase 1b clinical trial investigating the activity and tolerability of JMT101 (EGFR monoclonal antibody) plus osimertinib in EGFR 20ins-positive NSCLC. Based on matched patient samples before and after treatment, we also propose potential biomarkers for clinical benefits and putative resistance mechanisms in this population.

In the phase 1b trial, JMT101 6 mg/kg every 2 weeks plus osimertinib 160 mg daily yielded an IRC-assessed confirmed ORR of 36.4%, DCR of 95.0% and a median PFS of 8.2 months in the efficacy population ($n = 121$). The tumor shrinkage rate was 91.7%. Responses were observed in patients with different 20ins variants. In the subset of patients with platinum-refractory diseases who received investigated treatment in the second-line setting ($n = 53$), dual targeting EGFR 20ins with JMT101 and osimertinib led to a confirmed ORR of 34.0% with a median DOR of 13.3 months and a median PFS of 9.2 months. Currently, there are a number of emerging 20ins-directed therapies under development[5]. Most notably, CLN-081 reported a confirmed ORR of 39% and a median PFS of 12 months[21]. DZD9008, another novel EGFR TKI, also showed promising efficacy in a phase 1/2 study[22]. Poziotinib yielded a confirmed ORR of 46% in near-loop 20ins[20]. Generally, the antitumor activity observed with JMT101 plus osimertinib were comparable to amivantamab, mobocertinib and the above agents. The strength of this combination may lie in its CNS activity. More than half of the efficacy population in this study had baseline CNS metastases ($n = 62$), 80.6% among which were untreated. The confirmed ORRs were 33.9% and 39.0% in patients with and without baseline CNS metastases, respectively ($P = 0.559$). For the 16 patients with brain metastasis as target lesions, intracranial tumor shrinkage was observed in 81.3% of patients ($n = 13$). Intracranial DCR was 87.5%. The confirmed intracranial ORR was 25.0% (1 CR, 3 PRs). Intracranial CR was observed in one patient who progressed on poziotinib. Three patients had untreated leptomeningeal diseases at baseline. JMT101 plus osimertinib led to 1 PR (PFS = 14.2 m), 1 SD (PFS = 8.0 m) and 1 PD (PFS = 3.2 m) in these patients, respectively. The mechanism for CNS activity of JMT101 plus osimertinib is not fully clear yet. It may be attributed to high dose osimertinib, which also demonstrated desirable intracranial efficacy in patients with classic EGFR mutations[23,24]. Additionally, in vivo models of brain metastasis in breast cancer identified the heparin-binding epidermal growth factor-like growth factor (HBEGF) as a mediator of cancer cell passage through the blood–brain barrier[25]. The overexpression of EGFR and the activation of EGFR by HBEGF were reported to be involved in promoting brain metastasis of breast cancer[26,27]. If this mechanism also holds for EGFR 20ins-positive NSCLC, it could explain the CNS activity observed with JMT101 plus osimertinib in the present trial. Future studies are warranted to

elucidate the mechanism and key players of brain metastasis in NSCLC. Nevertheless, these data demonstrated that JMT101 plus osimertinib has the potential to be a new treatment option for patients with EGFR 20ins-positive NSCLC, especially for those with untreated CNS metastasis. Based on these data, a pivotal phase 2 trial (NCT05132777) is ongoing to support the regulatory approval of JMT101 plus osimertinib for EGFR 20ins-positive advanced NSCLC in China.

Osimertinib is an ATP-competitive covalent TKI designed against EGFR TKDs. Similar to other EGFR/pan-HER TKIs, its activity in 20ins is limited[28–30]. Increased dosage of osimertinib to 160 mg provided modest additional effects with a confirmed ORR of 25–28% and a median PFS of 6.8–9.7 months[15,17]. Responses to higher dose osimertinib in EGFR 20ins are still suboptimal. In a recent study, Elamin et al. reported that the insertion positions of 20ins could affect the drug-TKD interaction and determine tumor response to poziotinib[20]. As a TKI designed against EGFR TKDs, osimertinib, either 80 mg or 160 mg, may also be restricted by the same mechanism. In the present study, we attempt to overcome the restriction of TKD structure by targeting 20ins extracellularly and intracellularly at the same time. Responses to JMT101 plus osimertinib were seen across the spectrum of EGFR 20ins. In patients carrying helical, near-loop and far-loop insertions, confirmed ORRs were 75% (95%CI, 19.4–99.4), 36.7% (95%CI, 27.2–47.1) and 28.6% (95%CI, 8.4–58.1), respectively ($P = 0.297$). No concomitant EGFR alteration was identified to confer de novo unresponsiveness. Mechanistically, our preclinical study suggested that combining JMT101 with osimertinib led to a thorough and sustained EGFR blockade via signaling inhibition and receptor downregulation. The Fc-dependent, effector-mediated cytotoxicity of JMT101 and the increased EGFR availability on cell surface after osimertinib may also contribute to the antitumor activity of the combination[31,32]. Recent studies with electron microscope showed the allosteric connection between the extracellular domain and kinase domain of EGFR[33–35]. It was also reported that cetuximab could enhance the antitumor activity of an allosteric inhibitor (EAI045) by blocking EGFR dimerization[36]. The binding of JMT101 to the extracellular domain may also facilitate osimertinib binding to the kinase domain via dimerization blockade and conformational shift. Further studies are warranted to elucidate the mechanism of action for the combination. Nevertheless, data from this study indicate that dual targeting EGFR 20ins with JMT101 and osimertinib may bypass the limitations of TKIs and elicit tumor responses in a broader spectrum of 20ins variants.

Unlike classic EGFR mutations, there are few studies investigating tumor genomic features and their clinical implications for EGFR 20ins. Concurrent alterations in TP53, CDK4/6 and cell cycle-regulating genes have been identified as inferior prognostic factors for patients with classic EGFR mutations[37,38], while their clinical impacts in EGFR 20ins are largely unknown. In this study, we characterized tumor genomic landscapes in 142 patients with EGFR 20ins-positive NSCLC. The most common concurrent alterations in EGFR 20ins occurred in TP53 ($n = 60$, 49.6%), PIK3CA ($n = 10$, 8.3%) and RB1 ($n = 6$, 4.96%). TP53 comutations were found enriched in patients with early PFS events (<6 m, $P = 0.040$), but did not compromise initial treatment responses. In comparison to TP53 wildtype, TP53-altered tumors had significantly higher mutation load at baseline and throughout the course of treatment, which was consistent with its role in maintaining genomic stability. In general, the role of TP53 alterations in EGFR 20ins was similar to those in classic EGFR mutations[39]. The presence of TP53 alterations in EGFR 20ins did not affect initial treatment efficacy, but it might facilitate the development of acquired resistance by mediating genomic instability and cell cycle dysregulation.

Acquired resistance to EGFR targeted therapies in NSCLC predominantly fall into two categories, on-target (EGFR-dependent) resistance and off-target (EGFR-independent) resistance[40,41]. On-target resistance is mediated by acquired EGFR mutations that impede drug binding to TKDs. Off-target resistance is mediated by the shift of

oncogenic dependence that allow tumor to bypass EGFR blockade. At present, although resistance mechanisms to 20ins-targeted therapies are not fully characterized due to the lack of clinical data, we reckon that the resistance mechanisms in EGFR 20ins may partly overlap with those in classic EGFR mutations. Poziotinib is the compound that we currently have the most information on resistance. Preclinical studies and matched patient samples pre- and post-poziotinib showed that EGFR 20ins shared some common resistance mechanisms with classic EGFR mutations, such as T790M/C797S mutations, MET amplifications and epithelial-mesenchymal transition (EMT)[20]. In this study, we were able to obtain matched blood samples before treatment, on treatment and at disease progression from 38 patients. Serial cfDNA analysis identified potential resistance drivers in 42.1% ($n=16$) of patients, including PI3K/AKT activation, small-cell transformation, DNA repair defects and cell cycle dysregulation. Interestingly, unlike those observed in TKIs[20,41,42], no acquired EGFR alteration was detected in post-progression samples. Also, DNA repair defects and cell cycle dysregulation are more commonly implicated in resistance to chemotherapies rather than targeted therapies[19]. These findings suggest that targeting EGFR 20ins at the extracellular domain may lead to distinct patterns of resistance. Potential resistance mediators identified here need to be further validated in preclinical models. Hopefully, our findings could provide some insight into the characterization of resistance to other 20ins-directed therapies.

This study has several limitations. First, although we accrued a relatively large population of 150 patients, interpretation of the study outcome is limited by its early phase nature and a lack of control arm. Second, we used a targeted NGS panel, instead of whole-exosome sequencing, in cfDNA analysis. This may limit our identification of other potential resistance drivers. Additionally, no patient in this study had been treated with amivantamab or mobocertinib. It remains unclear whether this combination will be effective in patients progressing on the above two therapies.

Despite the above limitations, this study demonstrates that dual targeting EGFR 20ins with JMT101 and osimertinib has the potential to become a new treatment option for EGFR 20ins-positive NSCLC, especially for those with untreated baseline CNS metastases. Mechanistically, the combination led to a thorough and sustained EGFR blockade in distinct 20ins variants. Acquired resistance was predominantly driven by EGFR-independent mechanisms that partly overlapped with those observed in classic EGFR mutations. The ongoing pivotal phase 2 trial (NCT05132777) will further establish the activity and feasibility of JMT101 plus osimertinib in patients with EGFR 20ins-positive advanced NSCLC.

## Methods
### Participants
This study was conducted in accordance with Good Clinical Practice guidelines and the Declaration of Helsinki. The study protocol and all amendments were approved by institutional review boards at all participating sites, including Sun Yat-sen University Cancer Center, Fujian Cancer Hospital, Shanghai Chest Hospital, Zhejiang Cancer Hospital, Shanxi Provincial Cancer Hospital, West China Hospital, Hunan Cancer Hospital, Chinese PLA General Hospital, Henan Cancer Hospital, Union Hospital of Tongji Medical College, Hebei Tumor Hospital, Nanjing Drum Tower Hospital, Jiangsu Province Hospital of Chinese Medicine, Shanxi Bethune Hospital and Renmin Hospital of Wuhan University. All patients provided written informed consent before enrollment.

Eligible patients were aged ≥18 years, had histologically or cytologically confirmed stage IIIB or IV non-small-cell lung cancer (NSCLC) harboring EGFR exon 20 insertions (20ins), had at least one measurable lesion defined by Response Evaluation Criteria in Solid Tumors v.1.1 (RECIST v.1.1)[43] and Eastern Cooperative Oncology Group (EGOG) performance status of 0-1. Patients with asymptomatic brain or leptomeningeal metastasis were allowed. Key exclusion criteria included

prior treatment with anti-EGFR monoclonal antibodies, concurrent EGFR mutations that were reported to be responsive to approved EGFR tyrosine kinase inhibitor (TKI) (eg, exon 19 deletion, L858R, T790M, L861Q, G719X, S768I), use of immune checkpoint inhibitors within 3 months (for JMT101+Osimeritinb cohorts) and patients who had derived clinical benefits from previous EGFR-TKI treatments (CR, PR, or SD ≥ 6 months). Complete eligibility criteria are available in the study protocol (Supplementary Note).

### Study design and treatment
This was a multicenter, open-label, phase 1b, dose-escalation and dose-expansion study registered at ClinicalTrails.gov (NCT04448379, date of registration: June 25 2020). The dose-escalation stage followed a 3 + 3 design to assess the safety, tolerability and dosing of JMT101 in combination with afatinib or osimertinib in patients with advanced or metastatic NSCLC harboring EGFR 20ins. The first 6 patients were enrolled alternatively into cohort A1 (JMT101 6 mg/kg q2w + afatinib 30 mg qd) and cohort B1 (JMT101 6 mg/kg q2w + osimertinib 80 mg qd) to be evaluated for DLT for 28 days from the start of study treatment. If no DLT was observed, the next 6 patients would be enrolled alternatively into cohort A2 (JMT101 6 mg/kg q2w + afatinib 40 mg qd) or cohort B2 (JMT101 6 mg/kg q2w + osimertinib 120 mg qd). If a DLT was observed in the first 3 patients, the corresponding cohort would be expanded to 6 patients. If less than 2 out of 6 patients reported DLT in the cohort, the regimen would be considered tolerable. For cohorts that were considered tolerable, additional enrollment (≤ 12 patients in total) were allowed in the dose expansion stage. Safety and efficacy data were monitored periodically. One of the cohorts would be eventually selected for further expansion based on safety, tolerability and efficacy signals. Given the current treatment landscape of EGFR 20ins, the investigated therapy is required to have an ORR ≥ 35% to warrant its further development. Assuming that the expected ORR is 40%, when the sample size reach 120 in the efficacy population, the probability of observed ORR > 35% by normal approximation method is above 85% (86.8%). Patients were treated with JMT101 plus afatinib or osimertinib until disease progression, unacceptable toxicity, withdrawal of consent, or absence of further benefits judged by the investigator. Dose modifications or interruptions were allowed to manage toxicities. For each enrolled patient, EGFR 20ins was confirmed centrally by next-generation sequencing (HapOnco™ 107 panel) on formalin-fixed paraffine-embedded tumor tissue samples (preferred) or peripheral blood specimen collected at screening. Between June 29, 2020, and December 28, 2021, a total of 150 patients with EGFR 20ins-positive advanced NSCLC were enrolled into this phase 1b trial from 15 sites in China

### Endpoints and assessments
The primary objective was to evaluate safety and tolerability of JMT101 plus afatinib or osimertinib in advanced or metastatic NSCLC harboring EGFR 20ins. Secondary objectives included anti-tumor activity measured by tumor responses (ORR, DCR), duration of response (DOR), progression-free survival (PFS) and overall survival (OS), pharmacokinetics, immunogenicity, and biomarkers potentially associated with clinical outcomes.

Adverse events were monitored throughout the study until 30 days after the last dose, and were graded according to the National Cancer Institute Common Terminology Criteria for Adverse Events (NCI CTCAE), version 5.0. Disease assessment by radiologic imaging was conducted at screening, 4 weeks of study treatment and every 8 weeks thereafter. At screening, contract-enhanced CT scans of the chest, full abdomen and pelvis, contrast-enhanced brain MRI and a bone scan (ECT) are required for all participants (in case of allergy to the contrast medium or contraindication, plain CT scan or MRI scan is acceptable). For patients without baseline CNS metastasis, the following tumor assessment will include CT scans of the chest, full

abdomen and/or pelvis. Brain MRI will only be performed when clinical indicated. For patients with baseline CNS metastasis, the following assessment will include CT scans as mentioned above and a brain MRI. Tumor response was determined by a central independent review committee (IRC) and investigators per RECIST v1.1[44]. After disease progression, patients were followed up for survival every 8 weeks.

## Cell lines and cell viability assays

All mutant Ba/F3 cell lines were obtained from the KYinno BIO-TECHNOLOGY (https://www.kyinno.com/, #KC1050, #KC1025, #KC1024). All cell lines were maintained in RPMI1640 medium supplemented with 10% FBS and penicillin-streptomycin liquid in a humidified incubator with 5% $CO_2$. JMT101 were provided by Shanghai JMT-Bio Technology Co., Ltd. Afatinib (#S1011), osimertinib (#S7297) were purchased from SelleckChem.

Cell viability was determined using the Cell Counting Kit-8 assay (DOJINDO). Cells were collected from suspension medium, spun down at 300 g for 5 min and resuspended in fresh RPMI medium. 3000 cells per well were plated in 96-well plates. Cells were treated with ten different concentrations of inhibitors in serial threefold-diluted inhibitors or vehicle alone for a final volume of 100 μl per well. After 72 h, 10 μl of Cell Counting Kit-8 was added to each well. IC50 values were calculated using GraphPad Prism 9 at 50% inhibition. Each experiment was replicated three separate times.

## Antibodies and western blotting

For western blotting, EGFR mutant Ba/F3 cells were incubated with compound (10 ug/mL JMT101, 100 nmol/L EGFR TKIs or 100 nmol/L EGFR TKIs plus 10 ug/mL JMT101) for 6 hours and evaluating levels of pEGFR in cellular lysates using Western blotting for pEGFR, EGFR, pERK1/2, ERK1/2, pAKT, and AKT with antibodies from Cell Signaling Technology. Primary antibodies included rabbit anti-pEGFR (1:1000; Cell Signaling Technology, #3777 S, Rabbit monoclonal [D7A5], lot:16), rabbit anti-EGFR (1:1000; Cell Signaling Technology, #4267 S, Rabbit monoclonal [D38B1], lot:24), rabbit anti-pAKT (1:1000; Cell Signaling Technology, #9271 S, lot:15), rabbit anti-AKT(1:1000; Cell Signaling Technology, #4691 S, Rabbit monoclonal [C67E7], lot:28), rabbit anti-pERK (1:2000; Cell Signaling Technology, #4370 S, Rabbit monoclonal [D13.14.4E], lot:24), rabbit anti-ERK (1:1000; Cell Signaling Technology, #4695 S, Rabbit monoclonal [137F5], lot:28)]. Blots were probed with rabbit antibodies against GAPDH (1:10000; Proteintech, #10494-1-AP) as a loading control. Source data are provided in the Source Data file.

## Immunofluorescence staining

Ba/F3 cells after 24 h treatment (10ug/mL JMT101,100 nmol/L EGFR TKIs or 100 nmol/L EGFR TKIs plus 10 ug/mL JMT101) were seeded on poly-L-lysine coated slides. The slides were fixed in 4% formaldehyde for 15 min, permeabilized with Triton X-100 (Beyotime) for 10 min, and incubated with primary antibody overnight at 4 °C. The primary antibodies used in the study were rabbit anti-EGFR (1:100; Cell Signaling Technology, #4267 S, Rabbit monoclonal [D38B1], lot:24) at a dilution of 1:100. The slides were rinsed twice with PBS, followed by incubation with the appropriate Alexa Fluor 488-labeled Goat Anti-Rabbit IgG (Beyotime) for 1 h at room temperature. The cells were counterstained with Antifade Mounting Medium with DAPI (Beyotime), and the coverslips were mounted on slides.

## ADCC assays

In the ADCC assay, Ba/F3 cells overexpressing EGFR insASV, insSVD and insNPH were labeled with CellTrace Violet Cell Proliferation Kit (Invitrogen, #C34557) before pre-treated with 0.00001, 0.0001, 0.001, 0.01, 0.1, 1, 10, 100 μg/mL JMT101 (JMT-Bio Technology, # DP10720210902) or equivalent dose of Human IgG1, kappa Isotype

Control (SinoBiological, #HG1K, R1 clone, lot: MA16MY1804) for 30 min at 37 °C. NK cells were isolated from Peripheral Blood Mononuclear Cells, then co-cultured with three types of target cells at 37 °C (effector: target = 4:1) for 4 h. Cytotoxicity were analyzed by flow cytometry. Dilution methods for antibodies used in ADCC assays are provided in the Source data file.

## Flow cytometry

For ADCC assays, after NK cells were co-cultured with Ba/F3 cells for 4 hours, all cells were collected and rinsed by PBS. Cells were incubated by Propidium Iodide Staining Solution (BD,# 556463) for 15 minutes at room temperature. For EGFR expression on the plasma membrane, Ba/F3 cells after 6 h or 24 h treatment (10 ug/mL JMT101, 100 nmol/L EGFR TKIs or 100 nmol/L EGFR TKIs plus 10 ug/mL JMT101) were collected and rinsed by PBS. Cells resuspended by PBS were incubated by Alexa Fluor 488 anti-human EGFR (5ul/test; BioLegend, #352908, lot: B332782) for 15 minutes at room temperature. Alexa Fluor 488 Mouse IgG1, κ Isotype Ctrl (FC) Antibody (5ul/test; BioLegend, #400129, lot: B354284) were used as isotype control. Data were processed using FlowJo v10, BD FACSDiva Software v8.0.1 and CytExpert v2.4.

## In vivo studies

Female BALB/c nude mice at 6 to 8 weeks of age were obtained from Shanghai Lingchang Biotechnology Co. Ltd. The mice were housed in SPF-class independent ventilation cage (4 animals per cage). They were reared at 20–26 °C with a humidity of 40–70%, 12/12 dark/light cycles, and had free access to food and water ad libitum. Ba/F3 cells expressing EGFR A767_V769dup ($5 \times 10^5$ cells) were injected subcutaneously into the BALB/c nude mice. When the average tumor volume reached approximately 100 mm³, the mice were randomized to receive vehicle (po. qd), JMT101 (50 mg/kg iv. biw), afatinib (15 mg/kg po. qd), afatinib (15 mg/kg po. qd) + JMT101 (50 mg/kg iv. biw), osimertinib (25 mg/kg po. qd) and osimertinib (25 mg/kg po. qd) + JMT101 (50 mg/kg iv. biw). Tumor growth was monitored twice weekly by bilateral calliper measurements and weight of the mice was also recorded synchronously. The tumor volume was calculated using the formula 0.5 × (long diameter) × (short diameter)[2]. The animals were humanely sacrificed on day 14, and tumor tissues were harvested. The animal experiment was approved by the Sun Yat-sen University Cancer Center (SYSUCC) Animal Ethics Committee and handled in accordance with Good Animal Practices. The maximal tumor size permitted by the SYSUCC Animal Ethics Committee is 2000 mm³, which was not exceeded in the in vivo study except for one case. The tumor size of one mouse in the vehicle group exceeded the maximal size (2091 mm³) at the last measurement on day 14 due to rapid tumor growth. It was still below the maximal size (1087 mm³) on day 10. The mouse was humanely sacrificed once noticed.

## Pharmacokinetics

Blood samples for pharmacokinetic analysis were collected from patients to assess the plasma concentrations of JMT101 following a single dose and multiple doses (steady state) of JMT101. Plasma samples were obtained from patients at pre-dose, immediately after the end of the dose (+2 min), at 4 h (±15 min), 8 h (±30 min), 24 h (±1 h), 48 (± 2 h), 96 h (±4 h), 168 h (±7 h), and 240 h (±12 h) after the 1st and 3rd dose, as well as once before the 2nd and 4th dose. Plasma concentrations of JMT101 were determined using validated analytical methods. The pharmacokinetic parameters of JMT101 were derived using non-compartmental methods and comprised maximum plasma concentration (Cmax), time to Cmax (Tmax), area under the time-concentration curve (AUC), half-time (t½), Clearance (CL), etc.

Blood samples for immunogenicity tests were collected within 30 min prior to the prespecified doses (the 1st, 2nd, 3rd and 5th dose) and at the last visit (30 ± 3 days after the last dose). Samples were first

tested for anti-drug antibodies (ADA). Samples that were tested positive for ADA were further tested for neutralizing antibodies (Nabs).

## DNA extraction and sequencing

To explore potential biomarkers for response and resistance, patient blood samples were collected before treatment, on treatment (C2D1) and after disease progression. Blood samples were centrifuged to separate peripheral blood lymphocytes (PBLs) from plasma. Genomic DNA (gDNA) from PBLs was extracted using the RelaxGene Blood DNA System (TianGen Biotech Co., Ltd., Beijing, China). Cell-free DNA (cfDNA) and circulating tumor DNA(ctDNA) were extracted using the QIAamp Circulating Nucleic Acid Kit (Qiagen, Hilden, Germany). Isolated gDNA was sheared by dsDNA fragmentase (New England Biolabs, Ipswich, MA, USA). DNA fragments were selected based on size using AMPure XP beads (Beckman Coulter, Inc., Brea, CA, USA) and libraries were constructed using the KAPA Library Preparation Kit (Kapa Biosystems, Inc., Wilmington, MA, USA). Agencourt AMPure XP beads (Beckman Coulter, Inc.) were used for all clean-up steps. End repair and 3′-end A-tailing, PCR amplification, ligation and single-step size selection were performed following DNA fragmentation.

Targeted capture was performed using a custom set of biotinylated DNA probes, which cover 107 cancer-related genes (HapOncoCDx™, Roche). Amplified sample libraries and the SeqCap EZ Library were hybridized according to the manufacturer's protocol. Subsequently, the reactions were pooled and purified using Agencourt AMPure XP beads, and then amplified by PCR. After quantification by quantitative PCR, the library was diluted, denatured with 0.2 N NaOH, and sequenced using PE150 paired-end sequencing on the NovaSeq 6000 system (Illumina, Inc., San Diego, CA, USA).

DNA fragments, library purity and concentration were assessed using a Qubit 3.0 Fluorometer and dsDNA HS Assay kit (Invitrogen, Waltham, MA, United States). Fragment length was determined on a 4200 Bioanalyzer (Agilent Technologies, Santa Clara, CA, USA). cfDNA-seq was performed using a targeted next-generation sequencing panel on 107 lung cancer-related genes (HapOnco™ 107 panel) with a mean coverage of 2000×, and the mean coverage for gDNA is 1000×. Genes and alterations included in the HapOnco™ 107 panel are listed in the Supplementary Tab 5.

## DNA-seq data processing, alignment, somatic mutation calling and annotation

Raw cfDNA sequencing data were pre-processed by fastp v0.12.6 version 0.18.0 (https://github.com/OpenGene/fastp), which included adapter trimming, removing the reads in which the N base has reached a certain percentage (default length of 5 bp), removing reads which contain low quality bases (threshold value ≤ 20) above 40%, and sliding window trimming[43]. Clean reads were aligned to the hg19 genome (GRch37) using Burrows-Wheeler Aligner version 0.7.15-r1140 under default settings[45]. The Gencore version 0.12.0 (https://github.com/OpenGene/gencore) were used to remove duplicate reads[46]. Samtools version 0.1.19 (http://www.htslib.org/) was applied to generate pileup files for properly paired reads with mapping quality ≥60[47].

Somatic variants calling including point mutations, insertions and deletions was performed using VarScan2 version 2.3.8 (http://varscan.sourceforge.net/). The minimum read depth was 200, and the variant allele frequency (VAF) threshold was set at 0.1%[48]. Somatic variants (SNV or indel) that present in at least 5 unique reads, at least 1 on each strand with less than 0.5% mutant allelic frequency in the paired normal sample (gDNA from PBLs) were retained. A manual inspection was applied to further remove artifacts by GenomeBrowse® visualization tool (Version 2.x)[49]. Somatic mutation calls were annotated using ANNOVAR version 2018-04-16[50]. CNVkit version 0.9.3 (https://github.com/etal/cnvkit) was used for copy number variation detection[51], and

GeneFuse version v0.6.1 for structural variation detection (https://github.com/OpenGene/GeneFuse)[52].

## Statistics and reproducibility

This is an open-label, phase 1b, dose-escalation and dose-expansion study. No randomization or blinding was involved. Sample size was determined using the normal approximation method. Assuming that the expected ORR is 40%, when the sample size reach 120 in the efficacy population, the probability of observed ORR > 35% by normal approximation method is above 85% (86.8%). No data was excluded from the analyses. Safety and pharmacokinetics were evaluated in patients from all cohorts who received at least one dose of JMT101 plus afatinib or osimertinib during the dose-escalation stage and dose-expansion stage. The efficacy population included patients from the expanding cohort, either enrolled during the escalation stage or expansion stage, who had at least one dose of study treatment by the date of data cutoff (31 May, 2022). Target enrollment at the dose-expansion stage was 12–200 patients to allow preliminary estimation of antitumor activity. Categorical outcomes (eg, ORR, DCR) were presented as percentages and two-sided 95% confidence interval using the Clopper–Pearson method. Time-to-event outcomes (eg, PFS, OS) were estimated using the Kaplan–Meier method. Difference between categorical outcomes were tested using the Chi-square or Fisher's exact test. Survival curves were compared using the non-parametric log-rank test. Univariate and multivariate Cox regression analyses were performed to identify clinical and genetic factors associated with PFS. Subgroup analyses of efficacy by clinicopathological features and prior treatments were non-prespecified and exploratory. Tumor mutation load was calculated as the number of mutations detected in each sample. The distribution of tumor mutation load in different subsets of patients was presented as mean and standard error of mean. Mann-Whitney U test was used to compare the difference in distribution of tumor mutation load. Mutations that were newly detected or increased in VAF by 1.5 times at progression and defined as pathogenic mutations by OncoKB and COSMIC database were identified as potential resistance mediators that may drive acquired treatment resistance[53–55]. Potential mediators of resistance were subjected to KEGG pathway analysis[56]. If the patient harboured multiple oncogenic mutations that belonged to more than one pathways, he/she would be classified into the pathway with the highest number of mutations. If the same number of mutations were detected in more than one pathways, established mechanism of resistance reported for EGFR TKIs (eg, downstream activation, bypass activation) would take precedence. Statistical significance was defined as a two-sided P value < 0.05 or a hazard ratio excluding 1. Statistical analyses were performed using SPSS software (version 24.0.0 for Windows, IBM), SAS software (version 9.4) and R version 3.6.1 (http://cran.r-project.org).

## Reporting summary

Further information on research design is available in the Nature Portfolio Reporting Summary linked to this article.

# Data availability

The publicly available databases utilized for the biomarker analyses in this study include OncoKB (https://www.oncokb.org/)[53], COSMIC (https://cancer.sanger.ac.uk/cosmic)[54] and KEGG pathways (https://www.kegg.jp/kegg/pathway.html)[56]. The raw DNA-sequencing data generated in the study have been deposited in the China National Center for Bioinformation (http://bigd.big.ac.cn/) under the project number: PRJCA010856 (https://ngdc.cncb.ac.cn/gsa-human/browse/HRA002822). Sequencing or de-identified patient-level data are

available under restricted access. Access can be obtained by completing the application form via GSA-Human System (for sequencing data) and/or by contacting fangwf@sysucc.org.cn or zhangli@sysucc.org.cn. All requests will be reviewed by corresponding authors, the SYSUCC institutional review board and CSPC Pharmaceutical Group Co., Ltd. A signed data access agreement with the sponsors is required before data sharing. The complete protocol and statistical analysis plan are available in the Supplementary Note. The remaining data are available within the Article, Supplementary Information or Source Data file. Source data are deposited into Figshare (https://doi.org/10.6084/m9.figshare.22691635) and are provided with this paper. Source data are provided with this paper.

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

## Acknowledgements

The phase 1b clinical trial was sponsored by Shanghai JMT-Bio Technology Co., Ltd., a subsidiary of CSPC Pharmaceutical Group Co., Ltd. (CSPC). The sponsor provided the investigated drugs and worked with investigators on the trial design, data collection, data analyses, and results interpretation. Genomic sequencing was provided by HaploX Biotechnology Co., Ltd. We thank Bo Peng from HaploX for conducting the bioinformatic analysis. We thank Tingyuan Yang for providing assistance in manuscript writing and editing. The study was also supported, in part, by the National Natural Science Foundation of China (82241232 awarded to L.Z., 82002408 awarded to S.Z., 82173101 awarded to W.F.). We thank patients and their families for participating in the clinical trial. We also thank investigators and clinical research coordinators from all study sites.

## Author contributions

S.Z., L.Z., Y.H., and W.F. contributed to study design, data acquisition, interpretation and manuscript writing. S.Z., W.Z., B.H., Z.S., W.G., F.L., C.X., L.Z., Y.H., and W.F. contributed to patient enrollment, administration, data acquisition, and interpretation. L.W., Y.H., H.W., X.D., D.J., M.W., L.M., Q.W., J.Z., and Z.F. contributed to patient enrollment and administration. L.H. and Y.H. contributed to data acquisition and interpretation. L.L., R.H., Y.Y., M.L., and X.Y. contributed to data acquisition and technical support. All authors contributed to manuscript editing and approved the submission of the manuscript.

## Competing interests

L.H. is an employee of HaploX Biotechnology Co., Ltd. L.L., R.H., Y.Y., M.L., and X.Y. are employees of CSPC Pharmaceutical Group Co., Ltd. L.Z. has received research support from AstraZeneca, Eli Lilly, and Roche. The remaining authors have declared no competing interests.

## Additional information

[1]Department of Medical Oncology, State Key Laboratory of Oncology in South China, Collaborative Innovation Center for Cancer Medicine, Sun Yat-sen University Cancer Center, Guangzhou, China. [2]Department of Thoracic Oncology, Fujian Cancer Hospital, Fuzhou, China. [3]Department of Pulmonary Medicine, Shanghai Chest Hospital, Shanghai, China. [4]Department of Medical Oncology, Zhejiang Cancer Hospital, Hangzhou, China. [5]Department of Respiratory Medicine, Shanxi Provincial Cancer Hospital, Taiyuan, China. [6]Lung Cancer Center, West China School of Medicine and West China Hospital, Sichuan University, Chengdu, China. [7]Department of Thoracic Medicine, Hunan Cancer Hospital, Changsha, China. [8]Department of Medical Oncology, Chinese PLA General Hospital, Beijing, China. [9]Department of Medical Oncology, Henan Cancer Hospital, Zhengzhou, China. [10]Cancer Center, Union Hospital, Tongji Medical College, Huazhong University of Science and Technology, Wuhan, China. [11]Department of Medical Oncology, The Fourth Hospital of Hebei Medical University and Hebei Tumor Hospital, Shijiazhuang, China. [12]Department of Clinical Pharmacology, The Fourth Hospital of Hebei Medical University and Hebei Tumor Hospital, Shijiazhuang, China. [13]Department of Respiratory and Critical Care Medicine, Nanjing Drum Tower Hospital, the Affiliated Hospital of Nanjing University Medical School, Nanjing, China. [14]Department of Respiratory Medicine, Jiangsu Province Hospital of Chinese Medicine, Nanjing, China. [15]Department of Medical Oncology, Shanxi Bethune Hospital, Taiyuan, China. [16]Cancer Center, Renmin Hospital of Wuhan University, Wuhan, China. [17]Department of Respiratory Medicine, Jinling Hospital, Nanjing University School of Medicine, Nanjing, China. [18]HaploX Biotechnology Co,. Ltd., Shenzhen, China. [19]Clinical Science Division, CSPC Pharmaceutical Group Co., Ltd, Shijiazhuang, China. [20]These authors contributed equally: Shen Zhao, Wu Zhuang, Baohui Han, Zhengbo Song, Wei Guo, Feng Luo. [21]These authors jointly supervised this work: Xiugao Yang, Li Zhang, Yan Huang, Wenfeng Fang. ✉e-mail: yangxiugao@cspc.cn; zhangli@sysucc.org.cn; huangyan@sysucc.org.cn; fangwf@sysucc.org.cn

