## [Peer Review File · Nature Communications]

Reviewers' Comments:

Reviewer #1:

Remarks to the Author:

The authors have substantially improved the manuscript picking up the reviewer's suggestions. Nonetheless in spite of the significant changes and adds a subgroup analysis could help, a forest plot can facilitate at simple glance to see the benefit of the combination of osimertinib 160 mg plus JMT101.

Subgroup analysis of progression-free survival according to demographic characteristics and cluster of exon 20 EGFR mutations near or far loop and up-front co-mutations could be helpful.

What is the degree of benefit in comparison with poziotinib? Apparently, the response rate was lower for the near -loop mutations in comparison with that reported in the poziotinib study, 46% of response rate near the loop (Elain et al. Cancer cell 2022).

The study used JMT101 that requires coadministration of osimertinib (an ATP-competitive, EGFR inhibitor) however, the mechanism of such co-treatment is not explained in the manuscript. Is JMT101 as cetuximab disrupting asymmetric EGFR dimerization?

After receiving the revised manuscript,

I have read the comments and answers to my points, which I found all to be well detailed, complete and satisfactory. Therefore, as Reviewer 2, I am completely fine with the Authors' proper and careful reply.

Regarding Reviewer 4, in my opinion, all the queries raised by Reviewer 4 were very pertinent, positively punctilious. Again, the answers provided by the authors are very respectful and well addressed, fulfilling, in my opinion, all the issues. The authors have also very carefully indicated the changes included in the manuscript based on the reviewers' comments. The answers to Reviewer 4 certainly enhance the quality of the manuscript. I endorse the answers, since all of them explain very well and are easily understood. On the whole, the revised manuscript is very complete in all aspects. I agree that it may be published. I will be ready if you need any further clarification.

Reviewer #2:

Remarks to the Author:

thanks for submitting your article that has been extensively reviewed. I personally believe you have successfully addressed all the comments and I have no further comments

Reviewer #3:

Remarks to the Author:

Statistical analysis was appropriate. Sample size justification was provided for the expansion cohort (N=121) in the revision. Two statistical comments need clarification:

1. Dose escalation stage used 3+3 design for Cohort A1-A2, and Cohort B1-B2. However, it is unclear why Cohort A1 and Cohort B1 had n=12 which violates the 3+3 design (maximum n is 6).

While it is mentioned in the method section that "For cohorts that were considered tolerable, additional enrollment (≤ 12 patients in total)", the rule is not defined in the trial protocol. Moreover, with this rule (additional enrollment up to 12 patients), it is unclear how the 3+3 design was implemented. It may also raise ethical concern of putting patients in less effective dose cohort.

2. It is problematic to use PFS < 6 months vs ≥ 6 months as outcome for comparison in Figure 3 B-D. For patients with PFS censored < 6 months, they may have true PFS < 6 months or ≥ 6 months if they are followed up for a longer time. Please clarify how the PFS grouping was defined. For censored event, survival analysis will be more appropriate.

No.	Reviewer Comments	Responses
Reviewer 2		
1	The authors have substantially improved the manuscript picking up the reviewer’s suggestions. Nonetheless in spite of the significant changes and adds a subgroup analysis could help, a forest plot can facilitate at simple glance to see the benefit of the combination of osimertinib 160 mg plus JMT101. Subgroup analysis of progression-free survival according to demographic characteristics and cluster of exon 20 EGFR mutations near or far loop and up-front co-mutations could be helpful.	Based on your kind suggestions, we added a subgroup analysis of PFS based on demographic characteristics (gender, PS, line of therapy, CNS metastasis) and the type of 20ins to provide more information on the efficacy of JMT101 plus osimertinib 160 mg. A forest plot was added as Extended Data Fig 5b to make it more reader-friendly. Results (Page 13, Line 23): “Median PFS was similar in patients with different characteristics and different type of 20ins variants (Extended Data Fig 5b).”
2	What is the degree of benefit in comparison with poziotinib? Apparently, the response rate was lower for the near -loop mutations in comparison with that reported in the poziotinib study, 46% of response rate near the loop (Elain et al. Cancer cell 2022).	Thank you for your comments. Poziotinib did yield a higher ORR in near-loop 20ins than the combination reported in this study. However, the efficacy of poziotinib is restricted by the conformation heterogeneity of 20ins variants. The ORR in far-loop 20ins was 0%, while JMT101 plus osimertinib yielded an ORR of 28.6%. Meanwhile, the combination may also be effective in patients who progressed on poziotinib. One patient previously treated by poziotinib had a 25% decrease in overall tumor burden, CR in intracranial lesions and a PFS of 9.8 months under JMT101 plus osimertinib. Nevertheless, since both studies are single-arm exploratory trials, we tried to avoid cross-trial comparison in the manuscript. Therefore, we provided the above information in Results and Discussion to indicate that both treatments have the potential to become new treatment options for EGFR 20ins.

		Results (Page 13, Line 2): “One patient previously treated by poziotinib had a 25% decrease in overall tumor burden and a PFS of 9.8 months.” Results (Page 13, Line 13): “Intracranial CR was observed in one patient who progressed on poziotinib.” Discussion (Page 18, Line 7): “Poziotinib yielded a confirmed ORR of 46% in near-loop 20ins²⁰. Generally, the antitumor activity observed with JMT101 plus osimertinib were comparable to amivantamab, mobocertinib and the above agents.”
3	The study used JMT101 that requires coadministration of osimertinib (an ATP-competitive, EGFR inhibitor) however, the mechanism of such co-treatment is not explained in the manuscript. Is JMT101 as cetuximab disrupting asymmetric EGFR dimerization?	We appreciate your comments. Indeed, what you proposed is a highly potential mechanism of action for the combination. Jia et al. (Nature 2016) reported that cetuximab could enhance the antitumor activity of EAI045 by blocking EGFR dimerization. JMT101 could also disrupt EGFR dimerization. However, it is still unclear how this will affect the binding of osimertinib, since osimertinib is not an allosteric inhibitor like EAI045. Meanwhile, the dimerization dependence of different 20ins variants is also unknown. Structural and crystallographic studies are required to further elucidate their mechanism of action. Based on your comments, we revised the Discussion to further probe into the potential mechanism of action for the combination (Discussion, Page 20, Line 6): “It was also reported that cetuximab could enhance the antitumor activity of an allosteric inhibitor (EAI045) by blocking EGFR dimerization³⁶. The binding of JMT101 to the extracellular domain may also facilitate osimertinib binding to the kinase domain via dimerization blockade and conformational shift. Further studies are warranted to elucidate the mechanism of action for the combination.”

		The related article was also added in the Reference list: 36. Jia, Y. et al. Overcoming EGFR(T790M) and EGFR(C797S) resistance with mutant-selective allosteric inhibitors. Nature 534, 129-132, doi:10.1038/nature17960 (2016).
Reviewer 5		
1	Dose escalation stage used 3+3 design for Cohort A1-A2, and Cohort B1-B2. However, it is unclear why Cohort A1 and Cohort B1 had n=12 which violates the 3+3 design (maximum n is 6). While it is mentioned in the method section that “For cohorts that were considered tolerable, additional enrollment (≤ 12 patients in total)”, the rule is not defined in the trial protocol. Moreover, with this rule (additional enrollment up to 12 patients), it is unclear how the 3+3 design was implemented. It may also raise ethical concern of putting patients in less effective dose cohort.	Thank you for your comments. We apologize for not clarifying the process and rule for patient enrollment in the manuscript. Dose escalation stage used the 3+3 design. The rule that mentioned in the method section describes the dose expansion stage instead of the dose escalation stage. It corresponds to the following sentences in the trial protocol (Supp 1, Page 9): “All cohorts with good safety and tolerability in Stage I (up to 4 cohorts) are selected in which a certain number of additional subjects are included to further explore safety, tolerability, pharmacokinetic profile, and antitumor activity... Safety and efficacy data are monitored periodically during the course of the study. Considering the subject benefits, if a cohort shows clear evidence of treatment disadvantage, the enrollment into the cohort should be closed in advance to avoid more subjects receiving ineffective or low effective treatment; if a cohort shows clear evidence of treatment benefits, the recruiting number may be increased in that cohort.” The maximum number for additional enrollment was set based on the discussion of investigators to ensure reliable efficacy evaluation and timely determination of superior cohorts. Therefore, cohort A1, A2, B1 and B2 included patients from the dose-escalation stage and dose-expansion stage. The first 12 patients were enrolled alternatively into the four cohorts. No DLT was observed. Hence, these cohorts were deemed tolerable and additional patients were

		enrolled for further evaluation. It is indeed inaccurate to group all patients in cohort A1-B1 into dose escalation stage. Thank you for pointing out this issue for us. The following revisions were made in Results and Methods to correct this mistake. Results (Page 9, Line 5): “ Among them, 12 patients were enrolled during the dose-escalation stage, receiving JMT101 6mg/kg every 2 weeks plus afatinib 30mg daily (cohort A1, n=3), JMT101 6mg/kg every 2 weeks plus afatinib 40mg daily (cohort A2, n=3), JMT101 6mg/kg every 2 weeks plus osimertinib 80mg daily (cohort B1, n=3) and JMT101 6mg/kg every 2 weeks plus osimertinib 160mg daily (cohort B2, n=3), respectively. ... A total of 138 patients were enrolled into the dose-expansion stage (cohort A1=8, cohort A2=3, cohort B1=9, cohort B2=118) for further evaluation of efficacy and safety. Cohort B2 (JMT101 plus osimertinib 160mg) was selected for further expansion due to better efficacy-safety profiles observed and higher activity of osimertinib 160mg over 80mg shown in previous trials on EGFR 20ins.” The flow diagram was also revised accordingly to separate patients enrolled at different stages so that it could be more clear (Figure 1-R2). Methods (Page 27, Line 28): “For cohorts that were considered tolerable, additional enrollment (≤ 12 patients in total) were allowed in the dose expansion stage. Safety and efficacy data were monitored periodically. One of the cohorts would be eventually selected for further expansion based on safety, tolerability and efficacy signals.”
2	It is problematic to use PFS<6 months vs ≥ 6 months as outcome for comparison in Figure 3 B-D. For patients with PFS	We greatly appreciate your comments. In Fig 3B-D, we intended to identify potential biomarkers for clinical benefits. Therefore, patients

censored < 6 months, they may have true PFS < 6 months or ≥6 months if they are followed up for a longer time. Please clarify how the PFS grouping was defined. For censored event, survival analysis will be more appropriate.

were grouped based on the definition of “durable clinical benefit (DCB)”, which referred to patients achieving CR/PR/SD for at least 6 months (≥6 months). Patients with PFS censored < 6 months will be classified into the non-DCB group to ensure a more strict evaluation of clinical benefits. We should use “durable clinical benefit” instead of “PFS≥6 months” in the manuscript to avoid such confusion.

Based on your comments, we have revised **the X-axis in Fig 3B-D** and rewritten the sentences describing related results (**Results, Page 14, Line 21**): “To identify potential biomarkers for clinical benefits, baseline genomic profiles from patients with durable clinical benefits (DCB, defined as patients with CR/PR/SD for at least 6 months) were compared to those without. Concurrent TP53 alteration was found enriched in patients without DCB (P=0.040, Fig 3b).”

Results (Page 15, Line 9): “Further, A lack of durable clinical benefits also correlated with higher tumor mutation load at baseline (P=0.028, Fig 3c) and at C2D1 (P=0.021, Fig 3d).”

The legend for Figure 3 was also revised accordingly.

Reviewers' Comments:

Reviewer #1:

Remarks to the Author:

As per reviewer 2, the queried points have been correctly addressed.

Reviewer #3:

Remarks to the Author:

Statistical issues have been addressed.